



# Interdecadal Cycles in Australian Annual Rainfall

Tobias F. Selkirk[1], Andrew W. Western[1] and J. Angus Webb[1]

[1]Department of Infrastructure Engineering, University of Melbourne, Parkville, 3052, Australia

*Correspondence to*: Tobias F. Selkirk (selkirkt@unimelb.edu.au)

**Abstract.**

The extremes of Australian rainfall have profound economic, ecological and societal impacts; however, the current forecast horizon is limited to a few months. This study investigates interdecadal periodicity in annual rainfall records across eastern Australia. Wavelet analysis was conducted on rainfall data from 347 sites covering 130 years (1890-2020). Prominent cycles were extracted from each site and clustered using a Gaussian Mixture Model. This revealed three principal cycles centred

around 12.9, 20.4 and 29.1 years that were highly significant over red noise by t-test (p<0.0001). Overall, the three cycles combined had a mean contribution to total rainfall variance ($R^2$) of 13% across all sites, but this was up to 29% at individual sites. Both the 12.9-yr and 20.4-yr cycles were detected at over 95% of sites. The strength of each cycle varied over time and this amplitude modulation of the signal showed a systematic movement across the area investigated. 86% of extremely wet years fell within the positive phase of the combined reconstruction, with 80% of extremely dry years falling in the negative

phase. These results indicate underlying periodicity in annual rainfall across eastern Australia, with the potential to build this into long-term forecasts. This concept has been suggested in the past, but not rigorously tested. These findings open new paths for research into rainfall patterns in Australia and internationally. They also have broad implications for the management of water resources across all sectors.

## 1 Introduction

The peaks and troughs of Australian annual rainfall are known to be formidable (Nicholls et al., 1997). The highs can lead to extreme downpours like the 2022 floods that resulted in 5,000 uninhabitable homes and $6 billion in insured damages across eastern Australia (Deloitte Touche Tohmatsu Limited, 2023). The lows can extend for multiple years, as with the Millennium drought (2001-2009) - the true agricultural, economic and ecological impacts of which are still difficult to accurately quantify (van Dijk et al., 2013).

Are these extreme peaks and troughs random? Or could they be cyclical? Several researchers have searched for decadal patterns in precipitation with the goal of extending the limited forecast horizon of approximately three months (Hossain et al., 2018). Research over the past century into periodicity has predominately focussed on lunisolar influence (Currie & Vines, 1996; Noble & Vines, 1993), though more recent studies have moved towards alignment to climate drivers or a reluctance to name the source of the patterns observed (Rashid et al., 2015; Williams et al., 2021).





The lunisolar influence is generally thought to arise from either the 18.6-yr Lunar Nodal Cycle (LNC) or the ~11-yr sunspot cycle. Findings have been intriguing, with some studies claiming up to 85% of extreme flood and drought events in the world's major rivers occur in resonance with lunisolar cycles (Dai et al., 2019). Correlations have been found in South America (Currie, 1983), North America (Cook et al., 1997), China (Currie, 1995a), Mongolia (Davi et al., 2006), Egypt (Currie, 1995c) and Russia (Currie, 1995b). These studies have generally faced criticism for a lack of statistical rigor (Briffa,

1994; Burroughs, 2003) and failed to gather broad acceptance.

In Australia, it has been estimated that 19% of total rainfall variance can be attributed to the LNC and sunspot cycle (Currie & Vines, 1996). Noble and Vines (1993) used these two cycles to project forward annual rainfall in the Mallee (NSW) through to the end of the century, accurately predicting the Millennium drought seven years before it began. Follow-up studies on the state capitals (Vines et al., 2004), district averages in eastern Australia and the Southern Oscillation Index

(Vines, 2008) developed this concept further but lacked an objective statistical metric to support the findings.

The interdecadal signals in these earlier studies were isolated using a series of custom band-pass filters (Bowen, 1975) followed by Maximum Entropy Spectrum Analysis (MESA). The resultant waveforms showed both amplitude and frequency modulation, which is not uncommon in orbital drivers of natural phenomena such as the Milankovitch Cycles (Nisancioglu, 2009). However, matching to the underlying rainfall data required a phase change (i.e. inversion) of the 18.6-

yr LNC approximately every 100 years. Furthermore, the timing of this phase inversion was said to move slowly southward down the country (Currie & Vines, 1996). The authors attempted to visualise this migration using a series of polarity maps that tracked wet and dry alignments to the LNC peak, but they could provide no explanation for this phenomenon.

The lack of a plausible mechanism for the phase change was a challenge to the lunisolar interpretation of rainfall patterns and possibly a contributor to this work not being more influential. Influence of the LNC on rainfall was generally theorised

to act through additional gravitational pull on the oceans and atmosphere (Noble & Vines, 1993). Atmospheric tides are known to affect air pressure, which has subtle implications for wind fields, precipitation variations, thunderstorm frequency and temperature (Barbieri & Rampazzi, 2001). Calculations on solar and lunar tidal power have estimated they could account for more than half that required for vertical mixing in the ocean, with the upwelling of cold water periodically causing changes in Sea Surface Temperature (SST) in the Pacific Ocean. SST has an established relationship to rainfall in

eastern Australia (Keeling & Whorf, 1997; Treloar, 2002).

The relationship between cooling of SST in the eastern equatorial Pacific and rainfall in eastern Australia through the El Nino Southern Oscillation (ENSO) is well established (Bureau of Meteorology, 2021). Studies have also suggested that the 18.6-yr LNC has a predictive capacity for ENSO events extending back to AD 1704 (Yasuda, 2018). Kiem et al. (2003) proposed that ENSO was the dominant driver of flood risk in NSW, with some modulation from the Interdecadal Pacific

Oscillation (IPO). Later studies suggested that 80% of large floods across most of eastern Australia fall in 20 to 40 year cycles modulated by ENSO and IPO (McMahon & Kiem, 2018).

This paper revisits the concept of interdecadal cycles in eastern Australian annual rainfall with the aim of resolving some of these disparate earlier findings. A larger and more complete dataset was drawn on than in previous research, covering 130 years of gauged rainfall across 347 stations. Wavelet analysis is used to identify the variations in cycle amplitude and

frequency. Automated extraction of dominant frequencies allows for the analysis and aggregation of a large number of sites in the search for common cycles, freed from the assumptions of potential drivers proposed by the earlier research noted above. Moreover, clustering analysis was used to statistically verify the observed cycles, with three dominant cycles identified in the rainfall records. Visualising the spatial and temporal changes of these cycles across all sites provides an explanation for the previously described phase change.


## 2 Study Area and Data

A comprehensive list of 17,740 Australian weather stations was obtained from the Bureau of Meteorology Climate Data Online web portal (Australian Bureau of Meteorology, 2022). These stations were filtered to maximise length of record (>120 years, continuous up to 2022) and quality (>90% local record). This filtering was aimed at collecting the highest

quality gauged data to minimise uncertainty from using reconstructed rainfall. Few sites across western and central Australia met these criteria (<15% of total), driving the decision to focus on eastern Australia (latitude: -8° to -45° S, longitude: 133° to 155° E). This resulted in a list of 355 sites then used to access the infilled daily station data from 1889 to 2022 via the SILO database, hosted by the Queensland Department of Environment and Science (Jeffrey et al., 2001). Eight sites were not available from SILO, reducing the total down to 347. The daily rainfall data were summed by calendar year to give the total

annual rainfall for each station.

## 3 Materials and Methods

The method of analysis was broken down into three main sections to test specific characteristics (Fig. 1): a) identifying prominent cycles by cluster analysis of the wavelet global mean amplitude spectrum peaks over the full time-series, b) visualising the temporal and spatial variation of the dominant cycle by decade, and c) reconstructing the prominent cycles for

validation and assessment of contribution to rainfall.




### a) Identifying Prominant Cycles

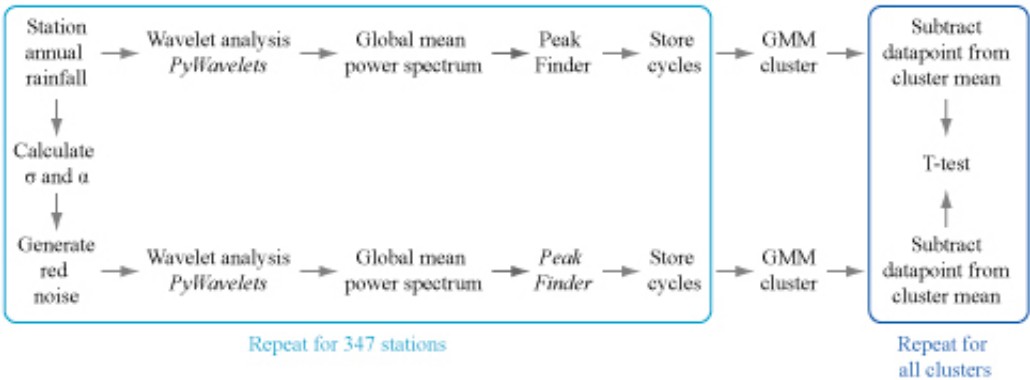

### b) Visualising Spatial and Temporal Changes of the Dominant Cycle

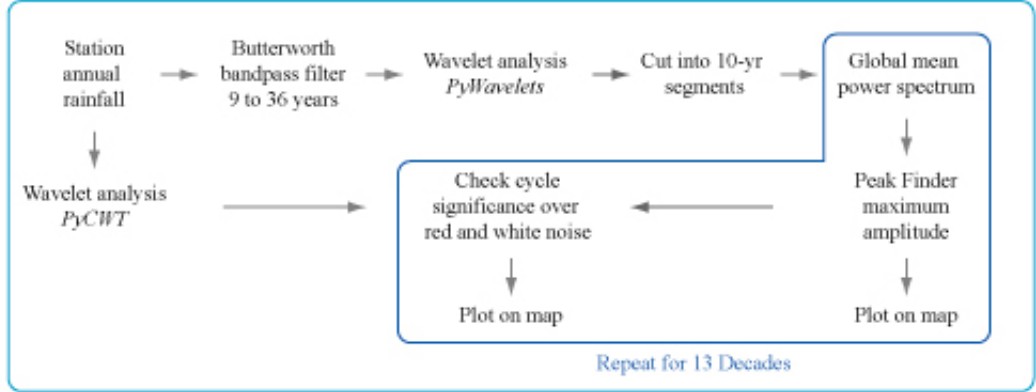

### c) Reconstruction of the Prominent Cycles and Contribution to Rainfall

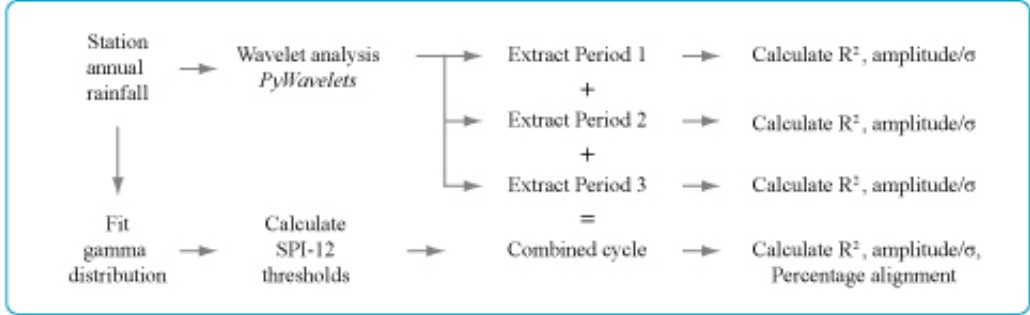

**Figure 1: A flow diagram summary of the method for each of the main sections.**



### 3.1 Identifying Prominent Cycles: Cluster Analysis of Global Wavelet Spectrum Peaks

Wavelet analysis was used to find all prominent cycles at each site across the full 130 year time series, and Gaussian Mixture Models (GMM) used to cluster the results across all 347 sites. Wavelet analysis is a time-frequency representation of a signal in the time domain. It has advantages over traditional wave decomposition techniques, such as Fourier Transform, in that it allows for the analysis of non-stationary processes and could therefore accommodate the frequency and magnitude modulation of cycles observed by previous researchers (Currie & Vines, 1996; Vines et al., 2004).

Three separate wavelet packages were used within the Python language, each with specific strengths. The *PyWavelets* package (Lee et al., 2019) was used for decomposing the signal, the *scaleogram* package for clear visualisation of the wavelet transform and *PyCWT* for significance testing of each extracted periodicity over white (random) and red (random with AR(1) autocorrelation) noise. In all instances, we used the standard *Morlet* wavelet function which is defined by Eq. (1) (Torrence & Compo, 1998):

$$\psi_0(\eta) = \pi^{-1/4} e^{i\omega_0\eta} e^{-\eta^2/2} \tag{1}$$

where $\eta$ is a dimensionless time parameter and $\omega_0$ is a non-dimensional frequency used to the eliminate units of measurement, generally taken as 6 (Farge, 1992). This essentially defines a complex plane sinusoidal function with a Gaussian envelope. *PyWavelets* allows for the parameterisation of wavelet bandwidth, which defines the width of this envelope in the time domain. A balance needs to be struck between time and frequency localisation, with wider bandwidths

providing increased accuracy of frequency estimation but lowered temporal resolution. In the initial extraction of cycles from the wavelet, a higher premium was placed on capturing more precise frequency values by setting the bandwidth to 6. We used a *continuous wavelet transform*, where a scaled and translated version of the function $\psi_0(\eta)$ is created by a discrete sequence of $x_n$ (Torrence & Compo, 1998):

$$W_n(s) = \sum_{n'=0}^{N-1} x_{n'} \psi * \left[\frac{(n'-n)\delta t}{s}\right] \tag{2}$$

where (*) is the complex conjugate. The wavelet spectrum is generated by varying the scale ($s$) and translation along the localised time index (n). This allows estimation and visualisation of how the amplitude and frequency of each component cycle in the raw data develop over the time series. For scale, we used the inverse of frequency to create cycle periods from 1 to 40 years, in 0.1-yr increments, to provide a clear resolution to the wavelet spectrum. Any finer resolution would have increased computation requirements without a noticeable improvement in accuracy.

The upper limit of 40 years was informed by the c*one of influence (COI)*. The wavelet function may extend past the limit of the analysed time-series as it approaches the boundary. This over-hang is generally padded with zeros, and the magnitude of



the spectrum within this region may be less reliable. For a *Morlet* function, the COI shortening at each end of the series is proportional to the period by a factor of √2 (Torrence & Compo, 1998). Accordingly, for a time-series of 130 years, a 40-yr period with will only achieve a single cycle within the COI and this can be seen as the practical upper limit of this technique

for a time-series of this length.

To automate the extraction of prominent cycles from the wavelet results, we generated a *global mean amplitude spectrum*. This was achieved by averaging the absolute value of the wavelet coefficients for each frequency scale over the full time series and plotting the magnitude against the cycle period. Averaging was used to smooth out variations and provide a clearer representation of the spectral data. We then used the *SciPy Signal* module *Find Peaks* to automate the listing of all

cycle periods found at that site (Virtanen et al., 2020). This process was repeated for each station to give a complete account of all cycles between 1 and 40 years in eastern Australia from 1889 to 2022.

We took this complete list of cycles and ran a Gaussian Mixture Model (GMM) to find significant clusters. We used the *Gaussian Mixture* object developed by *skikit-learn* (Pedregosa et al., 2011) to investigate which cycle periods were most prominent. GMMs are a form of unsupervised learning that assume that the dataset is a mixture of a finite number of

Gaussian distributions with unknown parameters, which can then be used to identify clusters. This approach has an advantage over "hard" clustering techniques in its ability to generate a probability distribution for each group. The package by *skikit-learn* also allowed for the optimum number of clusters to be independently determined by Bayesian Information Criteria (BIC) rather than having to be defined by the user. This BIC value was used to define the ideal number of clusters in the full list of cycles. The default package method of initializing the weight was kept (*kmeans*), but the number of random

seeds was increased to 500. GMMs can sometimes miss the globally optimal solution, thus a high number of random initialisations can help to overcome this issue (VanderPlas, 2016). The 0.1-yr resolution of the wavelet analysis is more precise than can be realistically achieved by this method, hence a Gaussian distribution of $\mu \pm 2\sigma$ was used to capture the possible range for each cluster.

To test the statistical significance of the clustering we used a standard t-test to compare each prominent cycle cluster with its

randomly generated equivalent. The *Statsmodels Time Series Analysis (TSA) Autoregressive Integrated Moving Average (ARIMA)* package was used to calculate the first order autoregressive AR(1) parameter ($\alpha$) and standard deviation ($\sigma$) for each station time series (Seabold & Perktold, 2010). This allowed for the generation of a custom red noise timeseries for every site. Lag-1 autocorrelation of annual rainfall over eastern Australia is relatively small ($\alpha < 0.13$) and not significantly different from zero at the 95 per cent confidence level (Simmonds & Hope, 1997). However, it can be higher at individual

sites and the use of red noise was thought to provide a more robust test. The wavelet analysis and peak extraction above was then repeated using the randomly generated data for all 347 sites, followed by GMM clustering.



The t-test was constructed to assess the central tendency of each cluster compared to red noise (Fig. 1a). The null hypothesis was that there would be no significant difference in the spread of periods between the two groups, indicating a random distribution of periods across all sites. The absolute residual was calculated by subtracting each data point (i.e. station) within a cluster from the cluster mean for both the annual rainfall data (group 1) and randomly generated data (group 2). We adopted a highly conservative type-1 error rate of 0.0001, given the length of the data set. Calculations were done using the *SciPy Statsttest_ind* module (Virtanen et al., 2020).

We further sought to control for the influence of any trend in rainfall. Strong trends can introduce harmonic distortion and artefacts into the higher frequencies of the wavelet spectrum analysis (Stéphane, 2009). Measuring the effect of climate change on annual rainfall in eastern Australia is complicated, but some regional and seasonal trends have been identified in recent years (CSIRO and Bureau of Meteorology, 2022). Despite the well-established trend in surface air temperature (Trewin et al., 2020), analysis of annual Australian rainfall suggests that there is no strong evidence of non-stationary rainfall for 90.5% of land area over the past century (Ukkola et al., 2019). To check for trend in the SILO rainfall stations used in this study, we applied an Augmented Dickey Fuller (ADF) test for stationarity. We repeated the wavelet analysis and GMM clustering having removed linear lest-squares fit trend from all station data using *SciPy detrend* (Virtanen et al., 2020) and compared the two sets of results.

### 3.2 Visualising Spatial and Temporal Changes of the Dominant Cycle

Previous authors have noted spatial and temporal variability in cycle influence which is reflected in amplitude modulation of the signal. For example, the timing of the 18.6-yr Lunar Nodal Cycle (LNC) "phase inversion" was said to move slowly south from 1860 to 1930 (Currie & Vines, 1996). To investigate this effect over time, the full wavelet spectra for each site were segmented into 10-year increments from 1890 to 2020. We used a Butterworth bandpass filter to focus solely on the range containing the significant cycles obtained from the GMM.

The Butterworth bandpass filter is commonly used in signal processing and climate analysis due to minimal distortion of the transmitted frequencies that pass through the filter (Roberts & Roberts, 1978; Sun & Yu, 2009). Selecting a wide band of cycle periods from 9 to 28 years with a gentle roll-off (order = 3) allowed us to repeat the test focussing solely on the range of frequencies under investigation and minimise artefacts.

Running wavelet analysis on the bandpass-filtered data, we generated a mean amplitude spectrum for each decade and once again used the *Find Peaks* module to extract the dominant cycle, defined as the peak with the highest magnitude. This allowed us to visualise the changes in all 347 wavelet spectra together over time by plotting the dominant cycle at each site in 13 time steps of 10 years over the full 130 year time frame.



Significance testing was also completed for each extracted cycle, over each decade, using the *PyCWT* package. The null hypothesis of this method assumes that the time series has a mean power spectrum defined by an AR(1) time series of 100 000 Gaussian white noise ($\alpha = 0$) or red noise ($\alpha > 0$) iterations. A peak in the normalised wavelet power spectrum that is significantly above this background level can be assumed to be a true feature (Torrence & Compo, 1998). For each site, the

wavelet spectrum was also generated using the *PyCWT* package, but in this case the unfiltered annual rainfall data were used as the input, with the default bandwidth of 1.5 providing higher time localisation accuracy. Each extracted cycle from the mean amplitude spectrum of the bandpass-filtered data was tested against these results for its significance over white and red noise. This method was developed to test the presence of each cycle independent of distortions introduced by trend, filtering, or individual package settings.

**3.3 Reconstruction of the Prominent Cycles and Contribution to Rainfall**

To quantify the influence of prominent cycles found using the GMM, we reconstructed the waveforms by extracting three fixed cycles from the wavelet transform of each site. The mean value of each significant GMM cluster was used rather than the individual peak extracted from the global mean amplitude spectrum at that site. This is because if we assume that the cycles each represent some fixed and enduring natural driver, then we are interested in the total influence of that specific

frequency at each site. The distribution observed for each cycle may represent variance in the accuracy of the wavelet transform relative to the noise, in which case the middle frequency is the best estimate. Wavelet analysis returns an array of coefficients corresponding to the continuous wavelet transform of the input signal for the frequency scales selected. We extracted the real component from this array for each selected frequency to visualise the waveform of that cycle over the full time series including the modulated amplitude.

We evaluated the coefficient of determination ($R^2$) between each cycle waveform and the annual rainfall anomaly. Since we are looking at cycles between 10 and 40 years the $R^2$ value may be excessively penalised by the large interannual variance of Australian rainfall. Therefore, as an additional measure of signal to noise, we recorded the maximum amplitude of the waveform and expressed this as a percentage of the standard deviation for each station (amplitude/$\sigma$). Adding together the timeseries of all three cycles gave a combined reconstruction, for which the $R^2$ was also calculated. To account for the edge

effects of the COI all $R^2$ values were calculated only between 1910 and 2000.

We sought to test the alignment of these cycles to years of extremely high or low rainfall at each station as a measure of their relationship to flood and drought. We used the Annual Standardised Precipitation Index (SPI-12) to characterise extreme rainfall years (World Meteorological Organization, 2023). The *gamma* package from *SciPy Stats* was used to calculate the shape, location and scale of a gamma distribution from the raw annual rainfall record at each site. This distribution was then

transformed to a normal distribution with a mean of zero, with values above or below ±2σ defined as extreme (Svoboda et al., 2012). This gave us a threshold value for extremely wet (SPI>2) or extremely dry years (SPI<-2) at each station. If the





annual rainfall exceeded either threshold value it was counted and compared to the phase of the combined reconstruction. For example, if an extremely dry year (SPI <-2) fell within the negative phase of the combined reconstruction it was said to be in alignment.

## 4 Results

The results are separated into the three major sections described in the methods. Section 4.1 explores the signals above those which could occur randomly. In Section 4.2 we investigate how the influence of these cycles has changed over time. Finally in Section 4.3, the prominent cycles are reconstructed to visually and quantitatively confirm the results from the previous sections.

### 4.1 Identifying Prominent Cycles: Cluster Analysis of Global Wavelet Spectrum Peaks

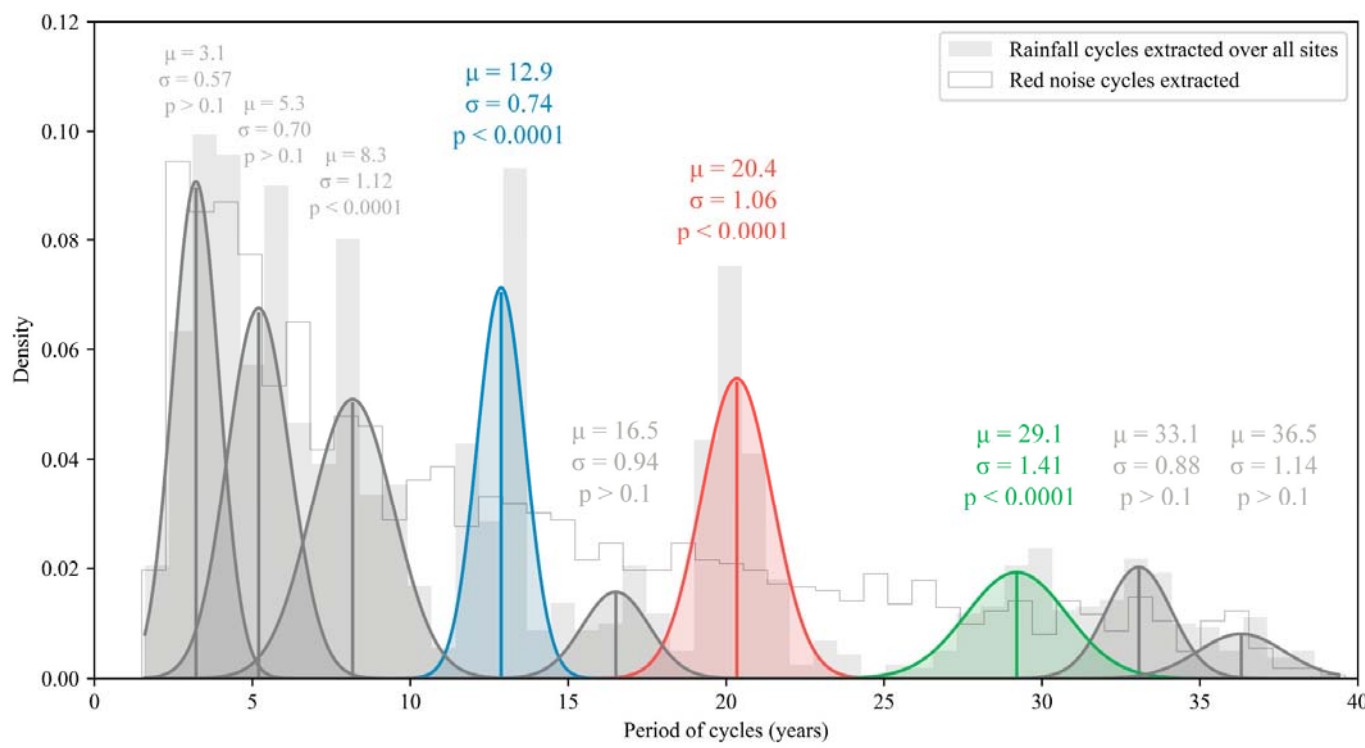

**Figure 2: Results for the extraction of cycles from 1 to 40 years using wavelet analysis for all 347 sites in eastern Australia. The filled light grey histogram represents the cumulative number of sites for each cycle length. The histogram outline shows the results of the same method using generated red noise. There are 9 distributions shown from the GMM with the mean (μ) and standard distribution (σ) for each cluster listed above. The p-value represents the results of t-test compared to red noise for each cluster. The three cycles centred at 12.9, 20.4 and 29.1 years that were highly significant are coloured in blue, red and green, respectively. The non-significant cluster distributions are coloured grey.**



The wavelet analysis revealed three cycles present at a large number of sites across eastern Australia (Fig. 2). These were centred (μ±2σ) around 12.9 (11.4 to 14.4-yr) present at 321 sites (93%), 20.4 (18.3 to 22.5-yr) present at 337 sites (97%) and

29.1 (26.3 to 31.9-yr) present at 104 sites (30%).

Another six clusters are present in the GMM results. A cycle at 8.3 years was also significantly different to red noise, but the high σ of 1.12 years relative to the mean would imply a wide range of periods unlikely to represent a single cycle (5.9 to 10.3 years). A cluster at 5.3 years was assumed to represent the ENSO oscillation, which shows quasi-periodicity between 2 and 7 years (Sarachik & Cane, 2010). This variation would result in a wider distribution as observed and is not significantly

different from red noise as it does not have a stable periodicity. Similarly the cluster at 3.1 years was indistinguishable from random noise. At longer time periods, there are three clusters between 25 and 40 years. The separation of the 33.1-yr and 36.5-yr centred clusters from red noise is not significant and the number of sites exhibiting the cycle is only slightly over what may be expected from random data.

Returning to the main clusters, the cluster at 29.1 years is highly significant although present at less than a third of sites, far

fewer than the ~13-yr and ~20-yr cycles. Despite its reduced prevalence it can still have a marked influence at sites where it is present (Fig. 3). In this individual wavelet spectrum, the magnitude of the ~30-yr cycle (green) suggests it has the strongest influence over the whole timeseries at this station. The global mean amplitude spectrum (Fig. 3d) shows that although the power is similar for the ~13-yr and ~20-yr cycles, the timing of their influence in the wavelet spectra (Fig. 3c) is staggered. From 1890 to 1950 the 19.5-yr cycle is dominant, but it then changes to the 12.4-yr cycle.

This changing of the dominance between the ~13-yr and ~20-yr cycles was observed in most of the individual wavelet spectra. The influence of the ~30-yr cycle tended to be more consistent across the timeframe observed, though this may be due to the length of the cycle within the timeframe observed and the COI. The example in Fig. 3 shows that the magnitude of the ~30-yr cycle can obscure the change in dominance between the other two cycles in the automated extraction. For this reason, along with its presence at a limited number of sites, we chose to exclude the ~30-yr cycle from the next phase of

analysis where we visualise the spatial and temporal changes of the ~13-yr and ~20-yr cycles.







**Figure 3: Sample of a single site analysis at Telegraph Point in NSW, SILO #60031, mean rainfall = 1314 mm/year. a) Annual rainfall anomaly timeseries with the linear trend (orange), b) station location, c) the wavelet spectrum up to 50 years showing three prominent cycles in the 10 to 40 year range with increasing power represented by shades of red. d) The cycle periods extracted from the global mean amplitude spectrum of the wavelet results, in this case the prominent cycles are 12.4-yr (blue), 19.5-yr (red) and 30.1-yr (green). The cycles are coloured equivalently to the GMM in Fig. 2. d) and e) Removal of the linear trend had little noticeable effect on cycle lengths, the 12.4-yr cycle lengthened slightly to 12.6 and the 39.9-yr cycle was no longer picked up by the Find Peaks program. All others remained unchanged.**






**4.2. Visualising Spatial and Temporal Changes of the Dominant Cycle**

Visualising the dominant cycle at each site by decade revealed a spatial pattern that developed systematically across the 130 year record (Fig. 4). Beginning with 1900-1910, we can see the ~13-yr cycle (purple) exerting a strong influence in the south west, with the ~20-yr cycle (red) dominating the eastern coastline. Fifty years later, from 1960-1970, the number of sites where the ~20-yr cycle is dominant has expanded to most of eastern Australia. As we continue to 2000-2010, the ~20-yr cycle is now dominant in the south-western part of the study area, and the ~13-yr cycle has begun to re-emerge along the eastern coast. This movement is best observed in the animation provided in Supplementary Material (S1). This reveals a southwest migration rate of approximately 140km/decade for the ~20-yr cycle.

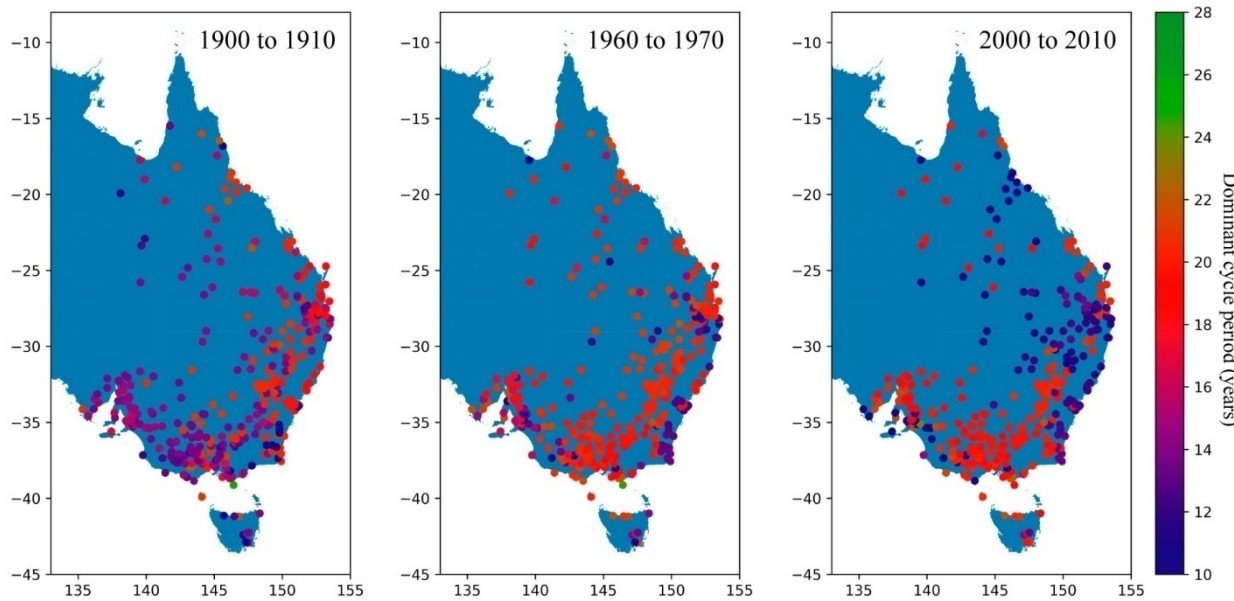

**Figure 4**: **Three selected decades showing the migration of the dominant cycle influence over 130 years. Each point represents the location of the analysed station, the colour defines the period of the dominant cycle in the 9 to 28 year range for that decade. The dominance of the ~20-yr cycle (red) moves southwest at a rate of ~140 km/decade. For an animation depicting the full set of decades see Supplementary Material (S1).**

The apparent movement of the dominant cycle results from amplitude modulation in both cycles (Fig. 5). Figure 5a shows the modulation of the ~20-yr cycle waveform isolated by bandpass filter, with its influence peaking around 1940-1980. The wavelet spectrum in Fig. 5d incorporates the significance test developed by Torrence and Compo (1998). Although the cycle is present over the entire timeseries, it is only significantly different to red noise between 1940 to 1980 (scale averaged power in Fig. 5d and the region circled in black in Fig. 5b).





**Figure 5**: **Sample of the *PyCWT* wavelet analysis of an individual site at Wilcannia in NSW (-31.5631, 143.3747), SILO #81000,**
**mean rainfall = 261 mm/year. Here we use two methods to illustrate the amplitude modulation of the ~20-yr cycle: a Butterworth**
**bandpass filter (a) and the scale averaged power of the wavelet analysis (e). Although the ~20 year cycle is consistent across the**
**whole timeseries, it only exceeds the 95% CI limit compared to a red noise spectrum from 1940-1980. a) The timeseries of the total**
**annual rainfall anomaly from 1890 to 2020 along with a Butterworth bandpass filter focussed around the ~20-yr cycle (18-23**
**years), which shows clear amplitude modulation. b) The wavelet spectrum with regions above the 95% confidence level circled in**
**black. c) The global power spectrum showing the significance cut-off levels for white (dashed pale grey line) and red (dashed black**
**line) noise. d) The scale averaged power in the same range used for the bandpass filter.**



Figure 6a gives a summary of the total number of sites where the dominant cycle is considered significantly different to white and red noise for each separate decade. At its peak in 1960 we can see that only a third of sites cross the 95% threshold for red noise. Figure 6b shows spatial coherence of these sites centred around the south of the mainland. This clustering of

significant sites can also be seen for the ~13-yr cycles at the beginning and end of the timeseries. For a full account of significant sites by decade see Supplementary Materials (S2).

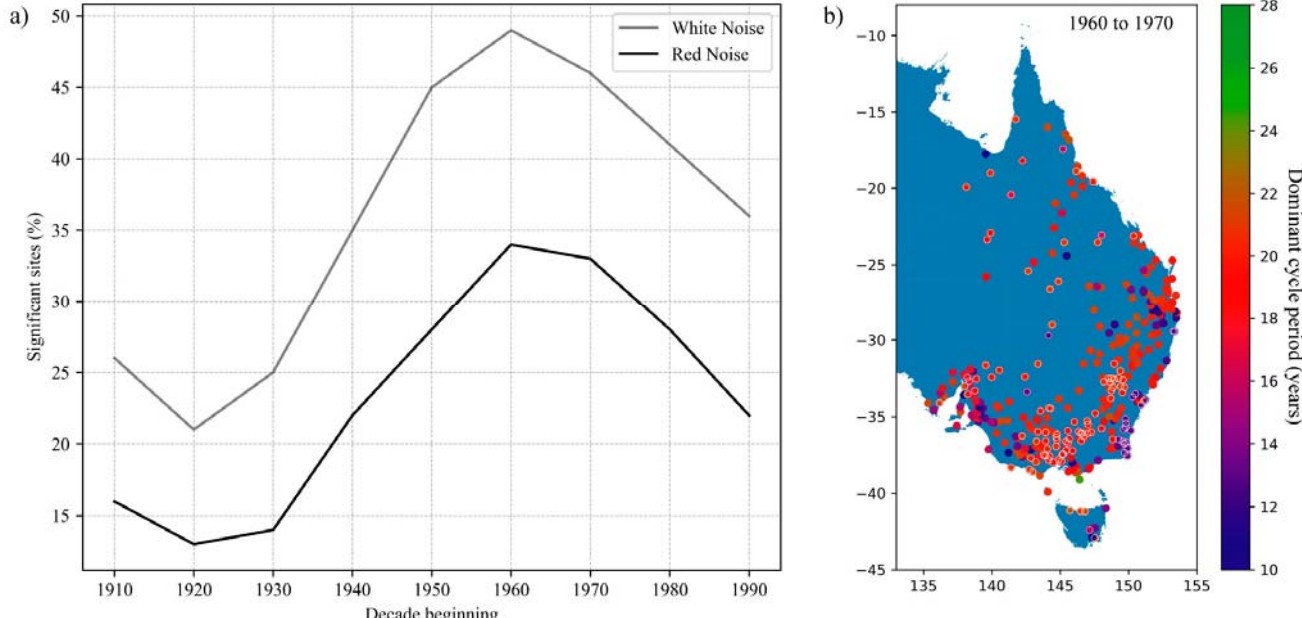

**Figure 6:Testing of significance for the dominant cycle at individual sites for each decade. a) The total percentage of sites where the power of the dominant cycle is over the 95% significance level (compared to red and white noise) for each decade from 1910**

**through to 1990 (truncated to account for COI). b) The sites circled in white are those with significant cycles over red noise. From 1960 to 1970 the dominant cycle between 9 and 28 years was significant at 34% of sites. See Supplementary Material (S2) for the full set of decadal maps.**



### 4.3 Reconstruction of the Prominent Cycles and Contribution to Rainfall

Extracting the three significant cycles found by GMM from the wavelet transform allowed us to visualise and quantify their influence on rainfall. The average rainfall variance ($R^2$) captured by the combined reconstruction across all sites was 13% (range 3 to 29%). The amplitude/σ ratio suggests a much stronger influence with the combined reconstruction having a mean of 126% (range: 55 to 224%).

Figure 7 shows a sample of the analysis carried out for each individual site. At this location, the combined reconstruction of all three cycles accounts for 9.4% of annual rainfall variance (Fig. 7a), less than the average $R^2$ across all sites**.** The ~20-yr cycle (Fig. 7e) was dominant at this site over the whole timeseries accounting for 8.0% of rainfall variance, and with a maximum amplitude reaching 83% of the site standard deviation (138mm). This is reflected in the wavelet spectrum (Fig. 7b), with a strong red band across the centre, and weaker pale blue bands at ~13 and ~30 years.

The annual rainfall anomaly crosses the threshold for extremely low rainfall (SPI < -2) five times, corresponding to major Australian droughts over the last century: 1937-1945, 1965-1968, 1982-1983 and 1997-2009. The troughs of the combined reconstruction align closely to each of these events, with all five years falling in the negative phase of the combined cycle (100%). Similarly, the threshold for extremely wet years (blue line) was crossed seven times, with nearly all falling in the positive phase (6/7 = 85.7%). The sole exception was the unusually high rainfall in 1939 when the combined cycles were approaching their nadir during the span of an extended drought from 1937 to 1945.

Looking at another reconstruction from the far north of Australia, we can also see a large trough in the combined reconstruction for the 1965-1969 drought (Fig. 8a). However, the magnitude of each cycle influence is markedly different here. The change in dominant cycle from the 12.8-yr to the 20.4-yrcyclecan be clearly seen occurring around 1970 (Fig. 8d-e) in agreement with the spatial pattern observed in Fig. 4. In this case, the 29.1-yr cycle has a much stronger and more consistent influence, accounting for 10.7% of rainfall variance and a large peak amplitude of 76% of standard deviation. The contrast in cycle magnitude and dominance between the two sites reflects divergent rainfall patterns, such as the extensive drought of 1937-1945 in the south (Fig. 7a), which did not occur in the north (Fig. 8a). Further examples of the full reconstruction for the largest (29%), mean (13%) and lowest (3%) $R^2$ can be found in the Supplementary Materials (S3, S4, S5).



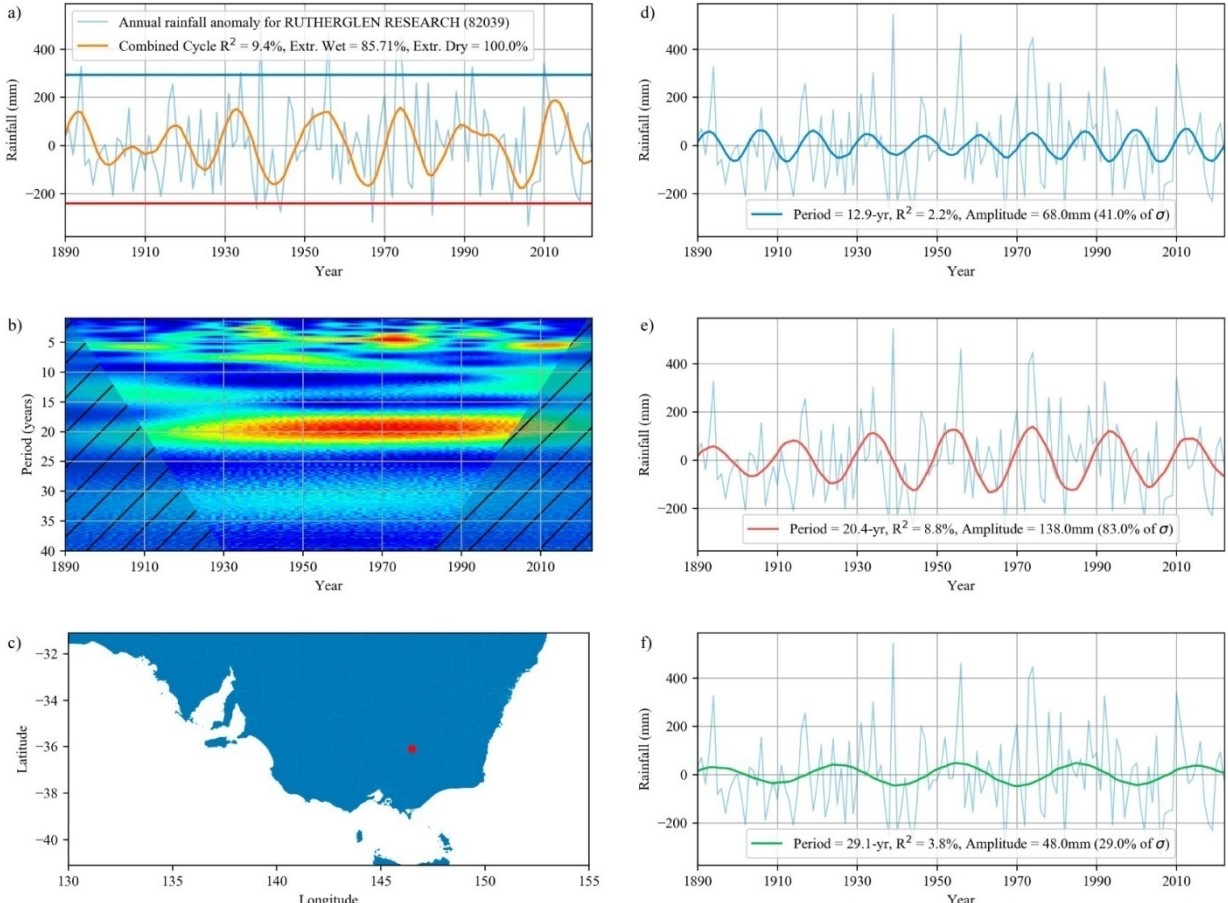

**Figure 7**: **An example of rainfall reconstruction by extracting the 12.9, 20.4 and 29.1 year cycles from wavelet analysis at Rutherglen Research station in Victoria (SILO #82039). a) the combined waveform (orange) of the three cycles, the contribution of each is shown in the column to the right (Fig. 7 d-f). The blue horizontal line represents the threshold SPI value for extremely wet years (>2), with dry years (<-2) in red. b) the wavelet spectrum for the site and c) its location. d-f) relative contributions of each cycle with values for R² and amplitude.**


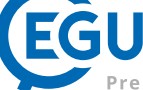

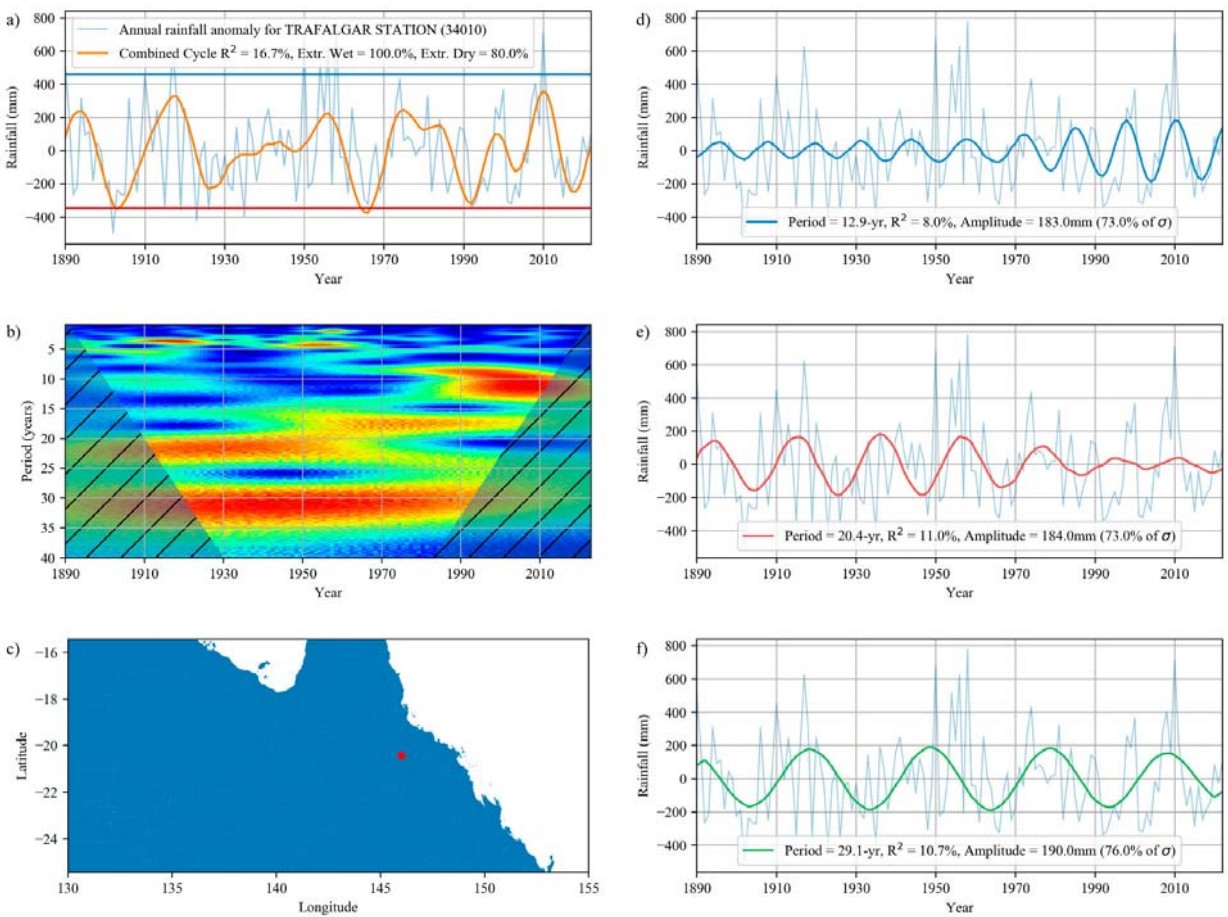

**Figure 8**: **An example of rainfall reconstruction by extracting the 12.9, 20.4 and 29.1 year cycles from wavelet analysis for an above average overall variance site at Trafalgar station in Queensland (SILO #34010). a) the combined waveform (orange) of the three cycles. The blue horizontal line represents the threshold SPI value for extremely wet years (>2), with dry years (<-2) in red. b) the wavelet spectrum for the site and c) its location. d-f) relative contributions of each cycle with values for $R^2$ and amplitude.**





Across all sites, the average percentage of extremely wet years that fell into the positive phase of the combined
reconstruction was 86%, and at 160 of the sites this alignment was 100%. The figures for extremely dry years were similar,
with an average of 80% across all sites, but with fewer in full alignment (74 sites).

The average rainfall variance ($R^2$) captured by the combined reconstruction across all sites was 13%, with the greatest
individual contribution coming from the 20.4-yr cycle (Fig. 9). The minimum $R^2$ for the 29.1-yr cycle is zero since it is not
found at all sites (Fig. 9c). The values for amplitude/σ hint at a greater influence than the coefficient of determination alone,
with the individual cycle scaling an average 43-68% of the standard deviation in rainfall. The maximum amplitude of the
combined reconstruction often greatly exceeds σ due to the constructive interference of the three cycles, such as in Fig. 7a
where the troughs of all cycles align around the time of Millennium drought (1997-2009).

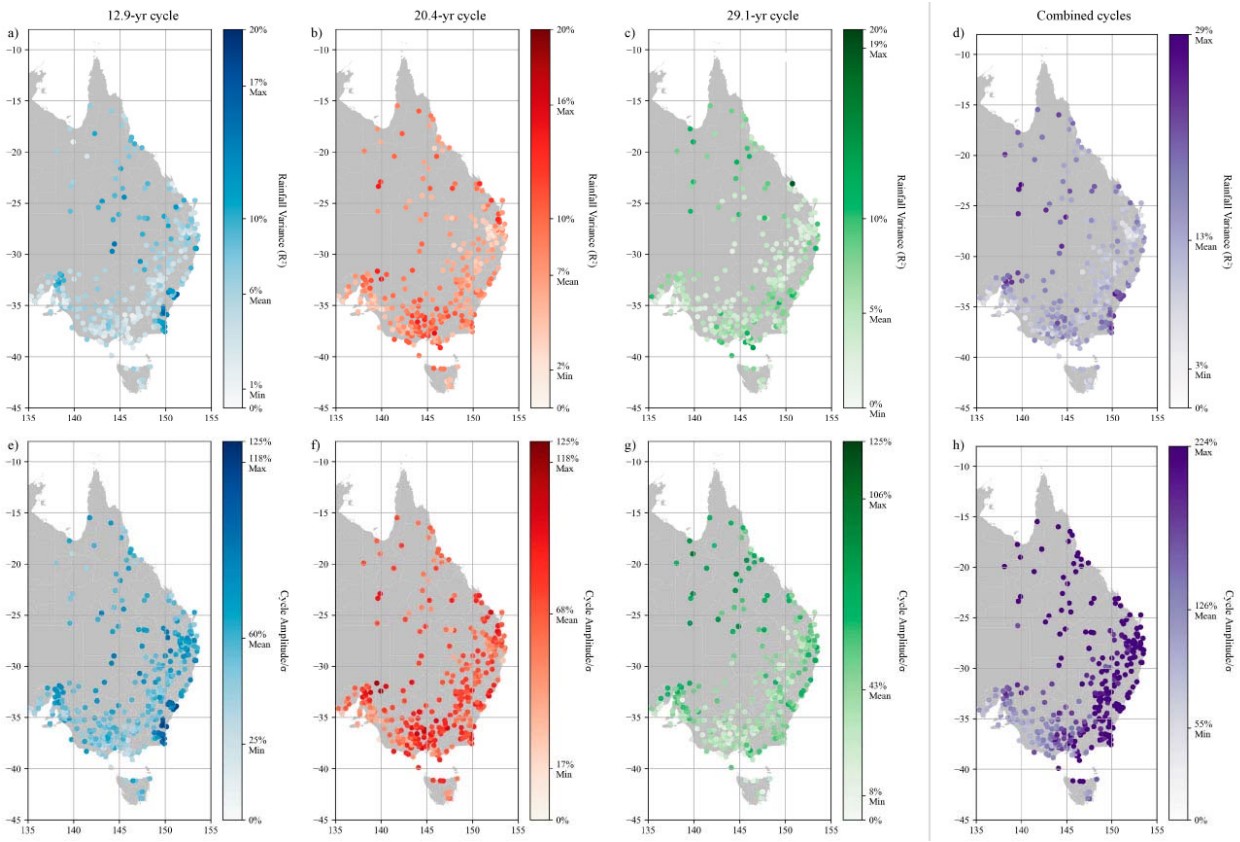

**Figure 9**: **Spatial distribution for the influence of each individual cycle as measured by the percentage variance of annual rainfall**
**and the cycle amplitude/σ across the entire timeseries. a-d) The colour bar represents the $R^2$ between the cycle extracted from**
**wavelet analysis and the annual rainfall anomaly for each site. e-h) The colour bar represents the peak amplitude of the extracted**
**waveform divided by the site standard deviation, expressed as a percentage.**





Figure 9 also shows how the contribution of each individual cycle varies by location. The influence of the 12.9-yr cycle is
particularly strong along the south-eastern coast. If we consider the results in Fig. 4, relating to amplitude modulation and
spatial variance, this is not surprising as we can see this region has kept a single dominant cycle for most of the last 130
years. The 20.4-yr cycle shows the strongest influence around central Victoria (lat-37, lon: 145), and has the most constant
contribution across all sites. The 29.1-yr cycle exhibits particularly high amplitude/σ in the northern regions, consistent with
the results of the individual analysis (Fig. 8f).

**5 Discussion and Conclusions**

This study found three cycles that are a prominent feature of temporal variability in annual rainfall in eastern Australia.
Furthermore, there appears to be a consistent spatial pattern evolving over time as a result of amplitude modulation. This
suggests the presence of external periodic drivers, and provides a significant advance upon previous findings.

Currie & Vines (1996) isolated a cycle centred at 18.3 ± 1.3 years in Australian rainfall, which they attributed to the 18.6-yr
Lunar Nodal Cycle (LNC). They used 308 stations with an average length of 84 years up to 1993, and excluded 15% of sites
where the cycle was not found. The rainfall data was also unfilled and not controlled for monitoring discontinuities. This
may account for the ~2-yr discrepancy with the current results, which used a longer and more complete dataset. Camuffo
(2001) suggested that the 18.6-yr LNC is not easily distinguished from the 19.9-yr Saturn-Jupiter cycle or the quasi-regular
22-yr Hale Magnetic cycle, all of which have been suggested to influence climate (Buis, 2020; Qu et al., 2012; Sorokhtin et
al., 2015). In Fig. 10e, we compare the alignment of the 18.6-yr LNC to the rainfall anomaly and the 20.4-yr cycle. The LNC
aligns well to rainfall in the early 1900s but has fallen out of phase by 2006.Overall, the $R^2$ of the LNC to rainfall across all
sites was <1%.

The 12.9-yr cycle found here is also slightly longer than the 10.5 ± 0.7 identified from 182 sites by Currie & Vines (1996)
and ascribed to the sunspot cycle. Similarly, if we compare the annual sunspot number to the rainfall anomaly (Fig. 10d) we
find little agreement at this site ($R^2$=0.01%), and across all sites there is essentially no correlation (mean $R^2$<0.5%).

The daily sunspot data was taken from the Royal Observatory of Belgium
(https://data.opendatasoft.com/explore/dataset/daily-sunspot-number@datastro/), averaged by calendar year, and scaled
equivalently to the extracted 12.9-yr cycle. This makes the previous attribution of the 18.6-yr LNC and ~11-yr sunspot cycle
as drivers of Australian rainfall unlikely, furthermore the 29.1-yr cycle cannot currently be accounted for by any currently
known lunar or solar driver.





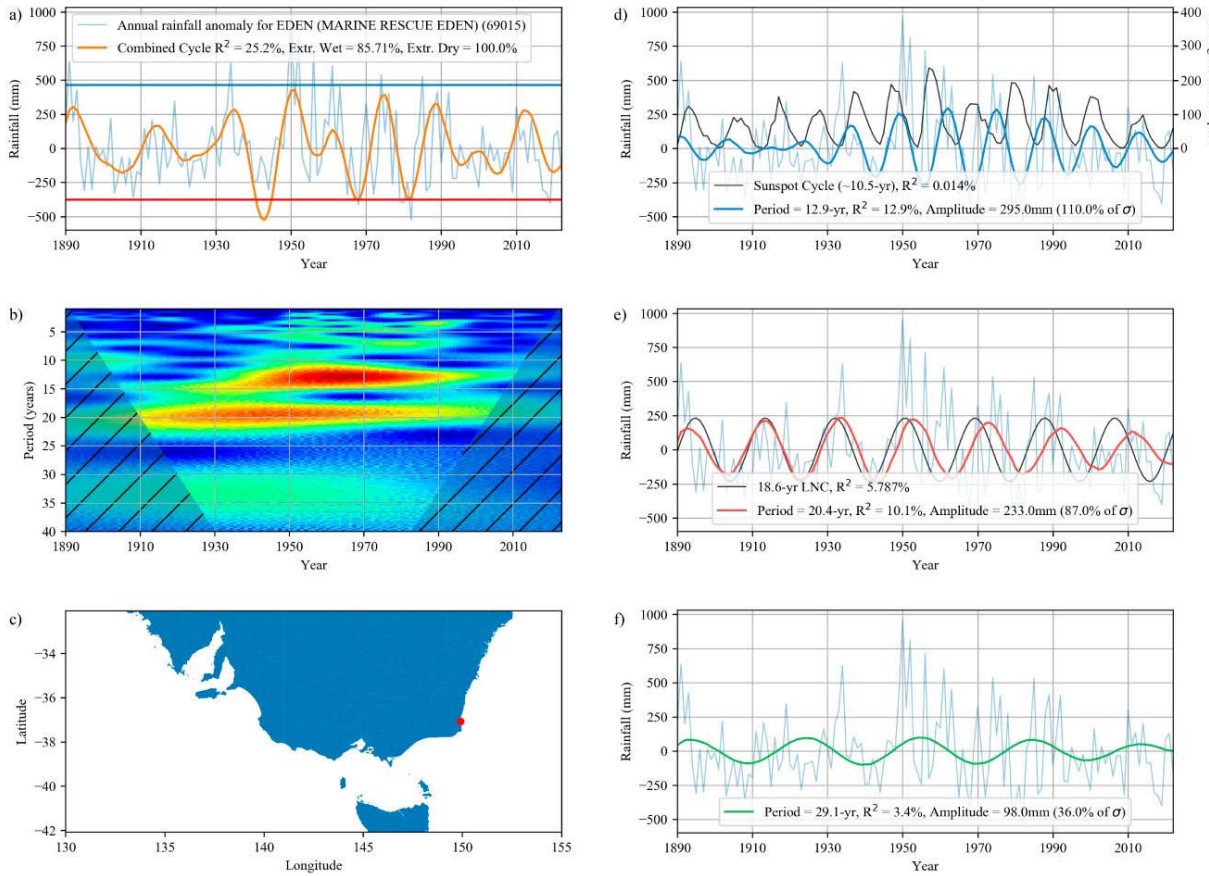

**Figure 10: An example of rainfall reconstruction by extracting the 12.9, 20.4 and 29.1 year cycles from wavelet analysis compared to the sunspot cycle and 18.6-yr LNC at Eden weather station (SILO #69015).The LNC was generated as a sinusoidal wave with**
**the peak aligned to the Major lunar standstill in March 1969 (Peng et al., 2019) and amplitude matched to the extracted cycle. a) The combined reconstruction of all three cycles from this study (orange), b) the wavelet spectrum, c) the station location. d) 12.9-yr cycle and rainfall anomaly have an R$^2$ of 12.9%, where as the average annual sunspot number (black) shows a very poor correlation to rainfall (0.014%). e) the 20.4-yr cycle and rainfall anomaly have an R$^2$ of 10.1%, the 18.6-yr LNC shows early agreement but steadily falls out of phase, with an R$^2$ of only 5.8%.**

The ability of wavelet analysis to view prominent cycles in the time domain, and map the changes spatially over decadal increments, allowed us to develop a coherent mechanism for what previous researchers had speculated to be a phase change. This showed that there is indeed a change that moves slowly southwest over time, though not an inversion of cycles, but an interchange of the dominant cycle. Figure 3c illustrates this change at a single site around 1950, with Fig. 4 showing the collective systematic movement of the ~20-yr cycle. The consistency of this pattern across 130 years and nearly all sites

makes it unlikely to be a result of the random non-linear and feedback mechanisms which govern weather patterns. However it also presents a challenge to many of the proposed mechanisms such as: seeding from cloud formation due to sunspot influence on air ionisation by cosmic rays (Fernandes et al., 2023), seeding of rainfall from cosmic dust (Adderley & Bowen,



1962), atmospheric tides (Kohyama & Wallace, 2016) and gravitational influence on SST via vertical mixing (Keeling &
Whorf, 1997). None of these would appear to allow for systematic change in dominant frequency by latitude and longitude
over 130 years.

The movement also does not seem to be affected by known climate drivers and climatic zones in eastern Australia. It moves
steadily through tropical, subtropical, grassland and temperate climatic zones. The influence of ENSO on rainfall is generally
consistent across most of eastern Australia (Tozer et al., 2023); in the north the monsoonal rainfall is associated with the
Madden-Julian Oscillation (Borowiak et al., 2023), and in the south the Southern Annular Mode has a marked effect on
climate (Meneghini et al., 2007). Whether the newly found cycles exist within any of these climate drivers is an avenue for
future research. Many studies have found periodic or quasi-periodic behaviour in the major climate drivers in the range of
18-22 years, again often attributed either the 18.6-yr LNC or the ~22-yr Hale Magnetic Cycle (Baker, 2008; Leamon et al.,
2021; Ormaza-González et al., 2022; Yasuda, 2009, 2018). The 2 to 7-yr quasi-periodicity of ENSO identified in our GMM
analysis is well known (Sarachik & Cane, 2010), and analysis of multi-proxy paleoclimate data dating back to 1650 has
identified a prominent 10-15 yr cycle in ENSO intensity noting that its causes are unknown (Sun & Yu, 2009).

Accurate paleoclimate data would be invaluable in testing the stability of these cycles over time however records in eastern
Australia are limited in locations and reconstruction skill is generally low (Flack et al., 2020). Reliable tree ring data is
scarce due to a lack of species with anatomically distinct annual growth rings (Heinrich et al., 2009), deep cave stalagmites
can provide extended timelines but must integrate rainfall over decadal scales (Ho et al., 2015), and coral luminescent
reconstructions from the Great Barrier Reef span up to 297 years, but only capture 47% of rainfall variability (Lough, 2011).
The lack of resolution and variance captured, along with the amplitude modulation of the cycles identified in this paper,
would mean that their impact could easily be hidden in spectral analysis. Williams et al., (2021) discovered a very similar
13-15-yr cycle in the Sierra Nevada district of the USA, accounting for 21% of precipitation variance over last century.
Extending the analysis back to 1400BC by tree ring reconstructions showed the cycle dipping in and out of significance by
wavelet analysis. The authors noted it was difficult to differentiate whether this was due to the amplitude modulation of the
cycle or the reconstructive skill of the record, though across the entire timeseries (1400-2020) spectral peaks at 12.8 and 21.3
years showed significant (90%) difference from white noise.

The two cycles identified by Williams et al. (2021) sit very close to the 12.9-yr and 20.4-yr cycles identified in this paper,
which signals the potential for using global rainfall data for validation rather than proxy paleoclimate data. Furthermore, the
behaviour over extended geographical regions may shed light on possible drivers of these cycles. Without greater
understanding of their origin and stability we must be wary of their potential use for forecasting. Although the cycles align to
over 80% of the peaks and troughs of extreme annual rainfall across all sites over the past 130 years, that is no guarantee that
they can currently be used to predict future flood and drought. Extracting the fixed cycles from each site for the



reconstruction gives us some confidence in their continuity over the period investigated. However, without a viable
mechanistic source of each cycle and its modulation we must proceed with caution.

Quantifying the scale of each cycle influence was also challenging. We sought to capture three dimensions in the metrics used: the coefficient of determination for overall variance, amplitude/σ as a measure of scale, and alignment to extreme rainfall for flood and drought. A limitation of using $R^2$ for this purpose is its sensitivity to outliers, meaning that irregular fluctuations in the rainfall may excessively penalise the correlation values and underestimate the cycle influence. It is for this
reason that the amplitude/σ metric was developed. At a single site the amplitude of the extracted signal would be given in mm/yr, however across the 347 sites there is a large difference in mean rainfall (150 to 2,319 mm/yr) and standard deviation (81 to 720 mm). Creating a ratio of the extracted signal amplitude to the total rainfall standard deviation the allowed comparison of the relative strength, comparable across all sites.

The concept that Australian rainfall may contain decadal cycles has been a source of debate for many decades. This study
has found evidence for the existence of the three interdecadal cycles in Australian annual rainfall centred around 12.9, 20.4 and 29.1 years. Together they account for an average of 13% (3-29) of rainfall variance across all sites. The combined amplitude of the waveforms over standard deviation is 126% (55-224) and they occur in alignment to over 80% of extremely high and low annual rainfall over the last 130 years. This degree of influence suggests they may be a valuable tool in long-term forecasting of rainfall in eastern Australia. Further research will focus on global datasets, climate indices, exploring
potential driver and viable mechanism of action.



**Code availability**

The following programs were used in the preparation of this manuscript:

Description: PyWavelets - Wavelet Transforms in Python

Citation in References: (Lee et al., 2019)

Software link: https://pywavelets.readthedocs.io/en/latest/

DOI: https://zenodo.org/records/13306773

Freely available under the Creative Commons Attribution 4.0 International licence.


Description: *scaleogram* - Wavelet Visualization Software

Citation in References: (Sauve, 2023)

Software link: https://pypi.org/project/scaleogram/

Repository: https://github.com/PyWavelets/pywt

Access Conditions: None

License: MIT License

Description: PyCWT - wavelet spectral analysis in Python

Citation in References: (Krieger & Freij, 2023)

Software link: https://pycwt.readthedocs.io/en/latest/

Repository: https://github.com/regeirk/pycwt.

Access Conditions: None

PyCWT is released under the open source 3-Clause BSD license

Description: Scikit-learn: Machine Learning in Python

Citation in References: (Pedregosa et al., 2011)

Software link: https://scikit-learn.org/stable/modules/clustering.html

Access Conditions: None

Copyright 2007 - 2024, scikit-learn developers (BSD License).


Description: Statsmodels: Econometric and Statistical Modeling with Python

Citation in References: (Seabold & Perktold, 2010)

DOI: 10.25080/Majora-92bf1922-011

Software link: https://scipy.org/install/



Access Conditions: None

Distributed under a liberal BSD license.

**Data availability**

The following data was used in the preparation of this manuscript:


Description: Australia-wide infilled daily rainfall station data

Repository Name: SILO - Australian climate data from 1889 to yesterday (Jeffrey et al., 2001)

DOI:10.1016/S1364-8152(01)00008-1

Website: https://www.longpaddock.qld.gov.au/silo/view-point-data/

Freely available under the Creative Commons Attribution 4.0 International licence.

Description: All Australian weather stations with length of record

Repository Name: Bureau of Meteorology - General station lists - list of all sites

Website: http://www.bom.gov.au/climate/data/lists_by_element/stations.txt

Freely available under the Creative Commons Attribution 4.0 International licence.

Description: Sunspot number

Repository Name: Daily total sunspot number since 1818 (taches solaires) - SILSO data, Royal Observatory of Belgium,
Brussels

Website:https://data.opendatasoft.com/explore/embed/dataset/daily-sunspot-number@datastro/table/?sort=-column_4

Freely available under the Creative Commons Attribution 4.0 International licence.

**Competing interests**

The authors declare that they have no conflict of interest.

**Acknowledgements**

This research was supported by an Australian Government Research Training Program (RTP) Scholarship.



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
