# Peer review of "Interdecadal Cycles in Australian Annual Rainfall"

_EGUsphere, 2024_

## Author Response (AR1)

**Author's response**

**Interdecadal Cycles in Australian Annual Rainfall**

Tobias F. Selkirk[1], Andrew W. Western[1] and J. Angus Webb[1]

[1]Department of Infrastructure Engineering, University of Melbourne, Parkville, 3052, Australia

*Notes:*

Reviewer comments are italicised.

Updated manuscript text is in quotation marks (" ")

**RC1**

*The manuscript is clearly written and presented. The methods used are standard and clear. The results are appropriately analyzed. The main shortcoming is that there is no effort to explain the results or to add insight to them. For instance, I assume that there are distinct seasons for rainfall in at least part of the region of Australia that was analyzed*

*Would it make sense to look for composites of atmospheric and oceanic state variables for the +ve and -ve phases of the 3 oscillations and/or to do a similar wavelet-clustering analysis of those variables and/or look for the wavelet coherence of some of those variables with the precipitation time series?*

*I think doing something like that would help make the paper more convincing*

*Some similar work is reported in https://www.mdpi.com/2306-5338/10/3/67*
* * *
The point raised about possible correlations to the dominant climate modes affecting rainfall in Australia and other oceanic-state variables is an excellent one and we have considered it at great length. Initial results were not included in this paper for several reasons but mostly due to the scope and report length:

- Initial aim and scope: the purpose of this study was to test the previous findings that the 18.6-yr Lunar Nodal Cycle and ~11-yr sunspot cycles were drivers of Australian rainfall using more comprehensive data and rigorous statistical analysis. The unexpected finding was that although the lunisolar cycles did not appear to be present there were other clear cycles of slightly different periodicity.
- Paper length: having discovered the cycles, more emphasis was put on a complete description and exhaustive statistical testing. Much of this work needed to be edited down to fit in a single paper and adding in wider climate variables would have involved either sacrificing some of the evidence required to first substantiate the claim, or increasing the length of the manuscript (which is already quite long). Neither of these seemed like viable options.

It is likely that these cycles are ultimately related to, or mediated by, known climate drivers in some way. Though the influence of the El Niño-Southern Oscillation (ENSO), the Indian Ocean Dipole (IOD) and the Southern Annular Mode (SAM) are significant in Eastern Australia, their impact is not limited to this region alone. We decided that the next step should be to first look at global datasets for other regions which may show similar evidence of influence by these cycles and then consider how this could relate to broader climate variables. Work on this front is currently in progress, and the manuscript was updated to reflect this rationale with the following text in the discussion (line 401, page 21):

"The movement also does not seem to be affected by known climate drivers and climatic zones in eastern Australia. It moves steadily through tropical, subtropical, grassland and temperate climatic zones. The influence of ENSO on rainfall is generally consistent across most of eastern Australia (Tozer et al., 2023); in the north the monsoonal rainfall is associated with the Madden-Julian Oscillation (Borowiak et al., 2023), and in the south the Southern Annular Mode has a marked effect on climate (Meneghini et al., 2007). Whether the newly found cycles exist within any of these climate drivers is an avenue for future research. Testing correlations of these established climatic modes known to affect Australian rainfall was not included in this study as it lay outside the initial scope and space required for a proper exploration. Given the unexpected finding that there were significant decadal cycles in Australian rainfall, more emphasis was placed on a complete description and exhaustive statistical testing. Moving toward this stage prematurely would have required sacrificing some of the evidence required to first substantiate the claim.

Many studies have found periodic or quasi-periodic behaviour in the major climate drivers of similar lengths to those described here. Cycles in the range of 18-22 years in ENSO and the SOI have been attributed to either the 18.6-yr LNC (Yasuda, 2009, 2018), or the ~22-yr Hale Magnetic Cycle (Baker, 2008; Leamon et al., 2021; Ormaza-González et al., 2022). Amonkar et al., (2023) found 5-7-yr and 12-14-yr cycles in Ohio River Basin streamflow were significantly correlated to ENSO. The 2 to 7-yr quasi-periodicity of ENSO identified in our GMM analysis is well known (Sarachik and Cane, 2010), and analysis of multi-proxy paleoclimate data dating back to 1650 has identified a prominent 10-15 yr cycle in ENSO intensity noting that its causes are unknown (Sun and Yu, 2009)."
* * *
**RC2**

*In this study, the authors investigate interdecadal periodicity in annual rainfall records across eastern Australia. They employ wavelet analysis and a Gaussian Mixture Model to extract and cluster prominent cycles from rainfall data collected at 347 sites covering 130 years (1890-2020). The results confirm the existence of an underlying periodicity in annual rainfall across eastern Australia, with three dominant cycles identified in the rainfall records. This analysis aims at explaining some of the disparate earlier findings and provides the basis for building long-term forecasts. This could open new paths for research into rainfall patterns in Australia and internationally, with broad implications for the management of water resources across all sectors.*

*The scientific contribution of this paper falls within the scope of Hydrology and Earth System Sciences. The paper is well-written with a clear and well-organized structure. The findings are appropriately discussed and related to previous works on the same topic. The discussion is efficiently supported by figures.*

*I suggest considering just some minor revisions:*

*While the concept of periodicity of rainfall events in Australia is widely and carefully explored, citing a number of previous studies, the choice of using wavelet analysis in this work is not adequately motivated. Is this the first application of wavelet analysis to a hydrological dataset? What are the advantages? Please provide a wider explanation and eventually cite previous papers using this technique to support your choice.*
* * *
This method is indeed very common in the exploration of periodicity in hydrological data, such as rainfall (Chowdhury & Beecham, 2012; Murumkar & Arya, 2014; Santos & Morais, 2013; Williams et al., 2021) and streamflow (Briciu, 2014; Brown et al., 2021; Gorodetskaya et al., 2024). Its main advantage over something like a Fourier transform is the ability to identify (and easily visualise) modulation in the period and amplitude of cycles - therefore accommodating some of the non-stationary properties observed by previous researchers (Vines et al, 2004). It was also the visualisation of cycle power in the time domain which allowed us to identify that was previously thought to be a "phase-change" was actually a change in the dominant cycle due to amplitude modulation. The manuscript was updated in the methods section as follows (line 90, page 5):

*"* Wavelet analysis was used to find all prominent cycles at each site across the full 130-year time series, and Gaussian Mixture Models (GMM) used to cluster the results across all 347 sites. Wavelet analysis is a time-frequency representation of a signal in the time domain and is frequently used in the exploration of periodicity in rainfall (Chowdhury and Beecham, 2012; Murumkar and Arya, 2014; Santos and Morais, 2013; Williams et al., 2021). It has advantages over traditional wave decomposition techniques, such as Fourier Transform, in that it allows for the analysis of non-stationary processes and could therefore accommodate the frequency and magnitude modulation of cycles observed by previous researchers (Currie and Vines, 1996; Vines et al., 2004). It is also computationally efficient and allows for the easy visualisation of cycle power across the spectrum, which can deepen the understanding of how certain cycles may be interacting to produce an observed effect."

What was novel about the way wavelet analysis was used in this paper was the automated peak extraction and cluster analysis by Gaussian Mixture Model. This was a newly developed method which allowed for the analysis of a much large number of spectra than usual, along with testing statistical significance of the number of sites compared to red and white noise. The results of wavelet analysis are often interpreted visually and independently in a limited region, such as Chowdhury & Beecham (2012)- 53 rain gauge stations, limited to South Australia, Murumkar& Arya (2014) - 4 rain gauge stations, limited to the Nira Catchment in India, or Amonkar et al., (2023) - 24 sites limited to the Ohio River Basin in America. This method allowed for the combined analysis of 347 rain gauge stations covering an area of nearly 4 million km2 - this is the largest area covered by any study that we are aware of into periodicity in rainfall using wavelet analysis. The discussion was updated with the following text to highlight this advantage (line 430, page 22):

"The two cycles identified by Williams et al. (2021) sit very close to the 12.9-yr and 20.4-yr cycles identified in this paper, which signals the potential for using global rainfall data for validation rather than proxy paleoclimate data. The novel method developed for this paper of automating peak extraction from each wavelet and clustering by GMM has the potential benefit of allowing for the processing of very large global datasets as opposed to the limited and regional areas for which it has traditionally been used (Chowdhury and Beecham, 2012; Murumkar and Arya, 2014)."
* * *
*The visualization of the results is crucial and I found all the figures suitable to convey the different messages about dominant cycles. The only figure that needs some substantial changes is the first. I suggest improving Fig.1 to make it easier to interpret. Furthermore, it appears to have a very low definition, I suggest improving the visualization quality.*
* * *
Thank you for the feedback on the figures. It is good to know that most were successful at conveying the message. Figure 1 was updated to include images of the separate stages to improve the clarity. The resolution of the image was also increased.
* * *
*In Fig.5, the letters that indicate the different panels do not correspond to the letters in the caption and in the text where the results are discussed.*
* * *
The text in Fig. 5 was updated to match the panels. The updated test reads:

"Figure 5: Sample of the PyCWT wavelet analysis of an individual site at Wilcannia in NSW (-31.5631, 143.3747), SILO #81000, mean rainfall = 261 mm/year. Here we use two methods to illustrate the amplitude modulation of the ~20-yr cycle: a Butterworth bandpass filter and the scale averaged power of the wavelet analysis. Although the ~20-year cycle is consistent across the whole timeseries, it only exceeds the 95% CI limit compared to a red noise spectrum from 1940-1980. a) The timeseries of the total annual rainfall anomaly from 1890 to 2020 along with a Butterworth bandpass filter focussed around the ~20-yr cycle (18-23 years), which shows clear amplitude modulation. b) the station location, c) The wavelet spectrum with regions above the 95% confidence level circled in black. d) The global power spectrum showing the significance cut-off levels for white (dashed pale grey line) and red (dashed black line) noise. e) The scale averaged power in the same range used for the bandpass filter."
* * *
*In Fig.4 and 6, I suggest changing the colormap to help the reader distinguish the cycle families and I also recommend making more evident the sites with significant cycles over red noise (I have difficulties in the identification of the points circled in white as it is now).*
* * *
The colormap of the landmass in Fig. 4 was swapped from blue to light grey to improve the visibility.

[Figure]

The colormap of the landmass in Fig. 6 was swapped from blue to light grey to improve the visibility and the significant cycles over red noise were circled in black as well as increasing the line width.

[Figure]

Figures 3, 5, 7 and 8 were also updated to feature the light grey landmass for readability and consistency. As well as the figures provided in Supplementary Materials.
* * *
*Page 5 line 102: possible typo "p"lane.*
* * *
Manuscript updated.
* * *
*Page 15 line 311: typo "20.4-yrcyclecan".*
* * *
Manuscript updated.
* * *
*Page 22 line 427: possible typo "the allowed".*
* * *
Manuscript updated, "the" deleted.
* * *
**Editor Feedback**

The references were reformatted to match the requirements of HESS as per the editors request.